# Human Fc-Conjugated Receptor Binding Domain-Based Recombinant Subunit Vaccines with Short Linker Induce Potent Neutralizing Antibodies against Multiple SARS-CoV-2 Variants

**DOI:** 10.3390/vaccines10091502

**Published:** 2022-09-08

**Authors:** Liqing Chen, Xiaoxiao Qi, Dan Liang, Guiqi Li, Xiaofang Peng, Xiaohui Li, Bixia Ke, Huanying Zheng, Zhongqiu Liu, Changwen Ke, Guochao Liao, Liang Liu, Qian Feng

**Affiliations:** 1Joint Laboratory for Translational Cancer Research of Chinese Medicine of the Ministry of Education of the People’s Republic of China, International Institute for Translational Chinese Medicine, Guangzhou University of Chinese Medicine, Guangzhou 510006, China; 2State Key Laboratory of Dampness Syndrome of Chinese Medicine, The Second Affiliated Hospital of Guangzhou University of Chinese Medicine, Guangzhou 510006, China; 3Guangdong Provincial Center for Disease Control and Prevention, Guangzhou 510006, China; 4Guangdong Hengda Biomedical Technology Co., Ltd., Guangzhou 510006, China; 5Guangzhou Laboratory, Guangzhou International Bio Island, Guangzhou 510006, China

**Keywords:** SARS-CoV-2 variants, spike protein, RBD, recombinant protein subunit vaccines

## Abstract

The coronavirus disease-19 (COVID-19) pandemic has been ongoing since December 2019, with more than 6.3 million deaths reported globally as of August 2022. Despite the success of several SARS-CoV-2 vaccines, the rise in variants, some of which are resistant to the effects of vaccination, highlights the need for a so-called pan-coronavirus (universal) vaccine. Here, we performed an immunogenicity comparison of prototype vaccines containing spike protein receptor-binding domain (RBD) residues 319–541, or spike protein regions S1, S2 and S fused to a histidine-tagged or human IgG1 Fc (hFC) fragment with either a longer (six residues) or shorter (three residues) linker. While all recombinant protein vaccines developed were effective in eliciting humoral immunity, the RBD-hFc vaccine was able to generate a potent neutralizing antibody response as well as a cellular immune response. We then compared the effects of recombinant protein length and linker size on immunogenicity in vivo. We found that a longer recombinant RBD protein (residues 319–583; RBD-Plus-hFc) containing a small alanine linker (AAA) was able to trigger long-lasting, high-titer neutralizing antibodies in mice. Finally, we evaluated cross-neutralization of wild-type and mutant RBD-Plus-hFc vaccines against wild-type, Alpha, Beta, Delta and Omicron SARS-CoV-2 variants. Significantly, at the same antigen dose, wild-type RBD-Plus-hFc immune sera induced broadly neutralizing antibodies against wild-type, Alpha, Beta, Delta and Omicron variants. Taken together, our findings provide valuable information for the continued development of recombinant protein-based SARS-CoV-2 vaccines and a basic foundation for booster vaccinations to avoid reinfection with SARS-CoV-2 variants.

## 1. Introduction

The COVID-19 pandemic has emerged as a result of the widespread infection of humans by the pathogen severe acute respiratory syndrome coronavirus-2 (SARS-CoV-2) [1,2]. The global pandemic has provided repeated opportunities for the single-stranded RNA virus to mutate. Indeed, variants continue to emerge and cause new outbreaks in various countries and regions. Up to now, more than 1000 SARS-CoV-2 variants have been detected; among these, the WHO has identified five “mutant strains of concern” that are posing severe challenges to epidemic prevention and control, including the Alpha, Beta, Gamma, Delta and Omicron variants [3,4,5,6]. The emergence of such variants, which can escape the immune response bolstered by existing vaccines and cause breakthrough infections [7,8], poses severe challenges to controlling the pandemic. Therefore, the development of a vaccine with broad spectrum protection against variants, including Omicron, is crucial.

SARS-CoV-2 is an enveloped virus possessing a positive-sense single-stranded RNA approximately 30 Kb in length, made up of spike (S), envelope (E), membrane (M) and nucleocapsid (N) sequences. The S protein is a class I fusion transmembrane structural glycoprotein composed of S1 and S2 subunits [9] and is responsible for the initiation of viral entry, pathogenesis and transmission. In particular, the receptor-binding domain (RBD) of S1 can recognize and bind to the host cell receptor ACE2 [4]. It has been proposed that inhibition of the RBD-ACE2 interaction may be useful in the prevention of SARS-CoV-2 infection [10,11]; thus, the SARS-CoV-2 RBD region is an attractive target for the development of new subunit vaccines.

Increasing amounts of evidence suggest the RBD as an immunogen that can trigger a potent functional antibody response to neutralize SARS-CoV-2 in vitro and in vivo by blocking the binding of the viral envelope protein to its host cell receptor ACE2 [12,13,14,15]. The apparent molecular weight of SARS-CoV-2 RBD (residues 319–541), is about 27 kDa. As a recombinant protein vaccine, SARS-CoV-2 RBD produces an insufficient immune response in vivo due to its small size and low immunogenicity. To solve this, the SARS-CoV-2 RBD recombinant protein vaccines use an RBD dimer or RBD trimer antigen to more effectively stimulate the immune system [16,17,18,19]. Several RBD-based SARS-CoV-2 vaccines are in development, with some already approved [20] and others in clinical trials [21,22].

Recombinant subunit protein vaccines have the advantages of a high yield, high safety, and easy storage and transportation, making them an important option for the prevention of COVID-19 [23]. Our recent study demonstrated that a murine mFc-RBD fusion protein recombination vaccine can improve recombinant immunogen solubility and stability and boost the vaccination-induced antibody response [24]. In this study, we fused the RBD to the human IgG1 Fc fragment (hFc) to boost the vaccination-induced antibody response. It is well known that the Fc fragment can serve as an immunopotentiator to enhance the cellular and humoral immune responses to the vaccine since it facilitates antigen delivery and presentation through interacting with Fc receptors on antigen presenting cells. In addition, the hFc-fusion can improve recombinant immunogen solubility and stability, and prolong immunogen half-life [25,26,27]. The direct fusion of functional proteins without a linker can lead to adverse outcomes, including protein misfolding, low protein expression, or impaired biological activity [28,29,30]. Thus, the rational design of an appropriate linker is critical to the construction of stable and bioactive fusion protein.

We first designed several fusion constructs using the same synthetic linker and evaluated the antibody and T-cell responses and the protective efficacy of these recombinant subunit protein vaccines. For comparison, vaccines based on recombinant protein with different linker sizes were also included. This is one of the first studies to directly compare how different linkers fusing the SARS-CoV-2 RBD might alter the immune responses in animal models. Furthermore, we designed both RBD-wild-type and RBD-mutant recombinant proteins to estimate the humoral and cell-mediated immune response to both wild-type and mutant strains of SARS-CoV-2 in mice. Our results lay a foundation for the development of vaccines against wild-type and mutated variants of SARS-CoV-2.

## 2. Materials and Methods

### 2.1. Cells, Reagents and Viruses

The Vero E6 cells and HEK293 cells were obtained from the American Type Culture Collection (ATCC) and cultured in Dulbecco’s Modified Eagle Medium (DMEM, Invitrogen, Waltham, MA, USA) supplemented with 10% fetal bovine serum (FBS), 10 mM HEPES, 1 mM sodium pyruvate, 1 × non-essential amino acids, and 100 U/mL of penicillin-streptomycin, pH 7.4. The SARS-CoV-2 strain 2019n-CoV/USA_WA1/2020 and SARS-CoV-2 variants including Alpha (B.1.1.7), Beta (501Y.V2), Gamma (P.1), Delta (B.1.617.2) and Omicron were obtained from the Guangdong Provincial Center for Disease Control and Prevention and Institute of Medical Laboratory Animals of the Chinese Academy of Medical Sciences. All experiments with infectious SARS-CoV-2 were performed in BSL3 facilities using protocols approved by the Institutional Biosafety Committee.

### 2.2. Animals

Five–six week-old specific-pathogen-free (SPF) female Balb/c mice were purchased from Southern Medical University (Guangzhou, China). Animal research was approved by the Guangzhou University of Chinese Medicine Animal Care and Use Committee (Ethics number: 2020W0007). All animals were given commercial mouse food and water ad libitum and housed in a temperature-controlled environment with a twelve-hour light–dark cycle.

All animals were immunized with recombinant protein vaccines or control via intramuscular injection in the hind leg. Two injections were given at days 0 and 30 (5 animals per group). All vaccine preparations were prepared using aluminum as an adjuvant, while the control group was injected with the same volume of PBS. Sera were collected before inoculation and at various time points after immunization for analysis.

### 2.3. Protein Expression and Purification

Sequences encoding the SARS-CoV-2 RBD, RBD-Plus, S1, S2 or S domain sequence fused with N-terminal hFc or His were inserted into the plasmid pcDNA3.1. Recombinant expression plasmids were transfected using the Expi293 Expression System (ThermoFisher Scientific, Waltham, MA, USA) according to manufacturer protocol. Supernatants were harvested on day 5 post-transfection. Recombinant proteins were purified using MabSelect SuRe LX Fast Flow and analyzed by SDS-PAGE. Briefly, 10% Tris–glycine SDS-PAGE was used to separate the proteins, and proteins in the gel were visualized by staining with Coomassie Brilliant Blue. The recombinant protein vaccines used in this paper have been provided by Guangdong Keguanda Biomedical Technology Co., Ltd. (Guangzhou, China).

### 2.4. ELISA Assay

An enzyme-linked immunosorbent assay (ELISA) was used to detect the specific antibodies in mouse antisera induced by recombinant RBD-hFc or RBD-His protein vaccines. Briefly, RBD-His or RBD-hFc protein was dissolved in carbonate buffer (0.1 M, pH 9.6) to obtain a 1 μg/mL solution. Ninety-six-well microtiter plates (ThermoFisher Scientific) were pre-coated with RBD-His or RBD-hFc (0.1 mL/well) overnight at 4 °C and blocked with 2% skim milk for 1 h at 37 °C. Diluted (1:3) mouse serum was added to the plates and incubated for 1 h at 37 °C, followed by 3 washes. Bound antibodies were incubated with goat anti-mouse HRP-IgG (1:2000), HRP-IgM (1:5000), HRP-IgG1 (1:2000), HRP-IgG2a (1:2000), HRP-IgG3 (1:2000) and HRP-IgG2b (1:2000) for 1 h at 37 °C. After washing, the substrate 3,3′,5′5′-tetramethylbenzidine (TMB) was added and the reaction was stopped by adding 1M H_2_SO_4_. Absorbance at 450 nm was acquired by an ELISA plate reader (PerkinElmer, Waltham, MA, USA).

### 2.5. Flow Cytometry

For recombinant proteins with the Fc domain, 293T cells were transfected with a vector containing human ACE2 (10 μg pCAG-hACE2/T75). At 48 h post-transfection, the cells were washed with PBS 3 times. The cells were harvested and seeded into an Eppendorf tube (1 × 10^6^/mL). An RBD-His protein (1 µg/mL) was added to the cells in the presence or absence of mice sera at indicated dilution (1:10). After incubation for 30 min at room temperature (RT), cells were rinsed with PBS. We then added the anti-6X His tag antibody-FITC (Abcam, ab1206, Cambridge, UK) at 1:100 dilution, and incubated at 4 °C for 30 min. For the positive control, RBD-His and anti-6X His tag antibody-FITC without antisera were added; for the negative control, neither pooled mice serum nor RBD-His was added. After thorough washing, the cells were resuspended in 1 mL PBS and fixed with 1% paraformaldehyde. The inhibitory effect was analyzed by flow cytometry (BD Biosciences, San Diego, CA, USA) and analyzed by FlowJo v10 software (FlowJo, LLC, Ashland, OR, USA).

For His-tag recombinant proteins, RBD-Fc followed by an anti-human IgG (Fc specific)-FITC conjugate (Sigma, 079M4801V, St. Louis, MO, USA) was used for analysis.

### 2.6. ELISpot

IFN-γ and IL-4 ELISpot assays were used to assess the T-cell response to the antigens. RPMI1640 without FBS was added to pre-coated 96-well plates to activate monoclonal antibodies (mAbs) against IFN-γ (Dakewe Biotech, 2210402, Shenzhen, China) and IL-4 (Dakewe Biotech, 2210005). We separated the spleens from vaccinated mice and harvested splenocytes. The splenocytes were suspended in RPMI 1640, plated in 96-well plates (1 × 10^6^ cells/well), and cultured with 10 μg/mL of RBD-Fc or RBD-His recombinant proteins corresponding to those of immunized mice at 37 °C and 5% CO_2_ for 24 h. The supernatants were discarded and cells were lysed in ddH_2_O at 4 °C for 10 min. After washing, biotinylated IFN-γ and IL-4 antibodies at 1:100 dilution were added into wells and incubated at 37 °C for an additional 1 h. After 6 washes, streptavidin-HRP (1:100) was added to the wells. After incubation at 37 °C for 1 h, 3-amino-9-ethylcarbazole was added to develop blots at 37 °C for 30 min in the darkness. The reaction was stopped with water and wells were air-dried. The IFN-γ and IL-4 spot-forming T cells were calculated using an ELISPOT Reader (Bio Reader 4000 Pro-X).

### 2.7. Neutralization Assay

Vero E6 cells were plated at a density of 2 × 10^4^ cells/well in a 96-well plate and grown at 37 °C overnight. Mouse sera were per-heated at 56 °C for 30 min. Serial 4-fold dilutions (1:4–1:1024) of mice sera were separately mixed with 100 TCID_50_ (50% tissue culture infective dose) of live SARS-CoV-2 virus or its variant and incubated at 37 °C for 2 h. Sera samples were tested twice. For each assay, cells infected with 100 TCID_50_ SARS-CoV-2 and 0.1 TCID_50_ SARS-CoV-2 were used, respectively, as positive and negative controls. The cytopathic effect (CPE) was observed daily and recorded after 72 h. Neutralizing titers of mice sera that suppressed 50% of the CPE were calculated by the Reed–Muench method.

### 2.8. Cell Surface Marker and Intracellular Cytokine Staining

One hundred μL of peripheral blood were collected from the mouse orbital venous plexus and anticoagulated with heparin sodium. Fluorescently-labeled cell surface molecular antibodies CD45 (PerCP), CD3 (FITC), CD4 (APC/Fire), CD8 (PE-Cy7) and CD44 (DAPI) were added and dark-incubated on ice for 20 min. After 3 washes with PBS, the cells were fixed with 150 μL of 4% paraformaldehyde and dark-incubated at RT for 20 min. A membrane-lysing agent was used to wash twice, followed by the addition of fluorescently-labeled intracellular antibodies IL-4 (APC) and IFN-γ (PE), and dark-incubation at RT for 20 min. The cells were washed, re-suspended and tested using a flow cytometer (BD FACSAria Ш, BD company, Franklin Lakes, NJ, USA). Data were analyzed using FlowJo V10 software (FlowJo, LLC, Ashland, OR, USA).

### 2.9. Statistical Analysis

All values were presented as mean ± SD. The statistical significance among different vaccination groups was compared using one-way ANOVA or *t*-test using Stata statistical software (GraphPad Prism 8, GraphPad Software Inc., San Diego, CA, USA). Values of *p* < 0.05 were considered significant.

## 3. Results

### 3.1. Construction, Expression, Purification, and Immunoassay of Recombinant Subunit Protein Vaccines

Here, we designed a series of recombined fusion protein vaccines encoding the full-length SARS-CoV-2 S protein (residues 1–1213), S1 domain (residues 1–685), S2 domain (residues 685–1213) and the RBD (residues 319–541) fused to either a His-tagged or human IgG1 Fc fragment with a 6-reside linker (ADDDDK) (Figure 1A). Briefly, pCMV3 fusion plasmids encoding either S-His, S1-His, S1-hFc, S2-His, RBD-His or RBD-hFc were transfected using the expi293 expression system (Figure 1B). The supernatant was collected after 3 days. Recombinant S-His, S1-His, S1-hFc, S2-His, RBD-His or RBD-hFc proteins were purified by column chromatography (Figure 1C) and visualized with an SDS-PAGE (Figure 1D). The purity of recombinant S-His, S1-His, S1-hFc, S2-His, RBD-His and RBD-hFc proteins was as high as 98%, which is important since high purity is critical in protein subunit vaccine development. We then tested the vaccines in female Balb/c mice and observed the immune response and production of neutralizing antibodies in (Figure 1E).

### 3.2. Recombinant Protein Vaccines with RBD Triggers Specific Powerful Humoral-Mediated Immune Responses in Mice

In order to detect the antibodies induced by recombinant subunit protein vaccines, IgG, IgM and IgG subclasses were analyzed at day 30 and 40 post-immunization. IgG is an abundant antibody and plays an important role in preventing infection. By 10 days after the second immunization, RBD-hFc triggered a robust humoral immune response in the mice, shown by an IgG titer up to 1.2 × 10^5^, higher than in the S1-hFc and S1-His groups (Figure 2A). An examination of the IgG levels showed that RBD-hFc could induce a significant titer by day 40, followed by S1-hFc and S1-His. The RBD-hFc and RBD-His also increased the IgG2b titer, while S1-hFc and S1-His increased the IgG2a titer. All vaccines had little effect on the IgG3 (Figure 2B). IgM is considered to be the most basic antibody produced by the B cells in response to foreign antigens. The IgM titers elicited by recombinant protein vaccines were significantly increased at 30 days post-vaccination, with the S1-hFc inducing the highest IgM titer, followed by the RBD-His and RBD-hFc (Figure 2C). Therefore, hFc-fused recombinant SARS-CoV-2 proteins exhibit strong antigenicity.

We next analyzed the neutralization activity of antisera collected from mice immunized with the different vaccines at days 30 and 40 post-immunization in Vero-E6 cells. As shown in Figure 2D, the mice vaccinated with RBD-hFc exhibited NA titers higher than those for S-His, S1-His, S1-hFc, S2-His and RBD-His (*p* < 0.001). These results show that RBD-hFc elicits a robust neutralizing antibody response against live SARS-CoV-2 infection in mice.

### 3.3. RBD-hFc and RBD-His Fusion Vaccines Can Block the Binding of RBD to ACE2

The spike protein of SARS-CoV-2 binds to ACE2 and facilitates the human-to-human transmission of the virus. In our previous research [24], we transfected HEK293T cells with ACE2, which we expected to maintain a native conformation, and then analyzed the RBD-binding activity by flow cytometry. As shown in Figure 3A, compared to the S1-hFc vaccine, the mice antisera from the RBD-hFc group at 1:10 strongly blocked RBD-His binding to ACE2, of which the inhibition rates were 73.48%. At the same time, the antisera from recombinant subunit protein with His-tag vaccines also can block RBD-hFc binding to ACE2. The order of inhibition efficiency is as follows: RBD-His > S1-His ≈ S2-His ≈S-His (Figure 3B). Taken together, our results suggest that the RBD recombinant protein vaccines, no matter fusion with hFc or His-tag, could induce the production of antibodies in mice to block the binding of SARS-CoV-2 to ACE2.

In the immune system, the gamma interferon (IFN-γ) and interleukin 4 (IL-4) levels are representative of helper T-cell (Th1 and Th2) activity. The IFN-γ/IL-4 ratio is an indicator of the Th1/Th2 ratio, which suggest the immune state of the body. In order to evaluate cellular immune function after the recombinant subunit protein vaccine injection, we isolated splenocytes from the immunized mice. Cellular immune responses, indicated by IFN-γ and IL-4 levels, were detected with an ELISPOT assay. As shown in Figure 3C and 3D, mice inoculated with RBD-hFc produced more IFN-γ spots (*p* < 0.001). Compared to the pre-immune serum, the numbers of IL-4 spots were significantly increased in both the RBD-hFc and RBD-His groups. The ration of IFN-γ/IL-4 in the RBD-hFc group was 0.37, suggesting that RBD-hFc induces humoral immunity.

### 3.4. RBD-hFc Fusion Vaccine Can Induce a Powerful Cell-Mediated Immune Response

Next, we wanted to know whether RBD-hFc vaccine immunization would activate specific T cells in vivo. CD44 is highly expressed by both the effector and memory T cells; we therefore analyzed the expression of IFN-γ and IL-4 in the CD44^Hi^^gh^ compartment of CD4^+^ and CD8^+^ T cells. We observed both the CD4^+^ (19.0% vs. 1.69%) and CD8^+^ (10.7% vs. 3.53%) T cells rapidly expand following RBD-hFc vaccination after 40 days compared to the control animals (Figure 4A,B). Notably, we consistently found that > 10 times more IFN-γ than IL-4 was produced by the CD4^+^ T cells stimulated with RBD-hFc (Figure 4A). Like the CD4^+^ T cells, the CD8^+^ T cell production of IFN-γ and IL-4 increased following the vaccination (Figure 4B). These results demonstrate that the RBD-hFc vaccine can effectively induce both robust systemic humoral and cell-mediated immune responses in mice.

### 3.5. Wild-Type SARS-CoV-2 RBD-Plus-hFc with a Small Linker Can Elicit a Strong Humoral Immune Response

We previously found that a SARS-CoV-2 RBD hFc fusion protein vaccine can produce strong humoral and cellular immunity in mice. Compared to smaller antigens, larger protein antigens engage more efficiently in interactions with B cell receptors, thereby facilitating the generation of high-affinity antibodies. Therefore, to most effectively stimulate the immune system, we designed a wild-type SARS-CoV-2 RBD-hFc (residues 319–541; “W-3”) and RBD-Plus-hFc (residues 319–583; “W-1”) recombinant to immunize mice and assess their immunological responses and protection against SARS-CoV-2. We found that, at the same antigen dose, serum from the W-1 mice showed higher IgG (Figure 5B) and IgM (Figure 5C) antibody titers and live virus-neutralizing antibody titers (Figure 5D) compared to serum from the RBD-hFc mice. This suggests that the RBD-Plus-hFc fusion protein is more antigenic than the RBD-hFc protein.

Linker length can directly affect the biological activity of a fusion protein vaccine. We designed a series of vaccines encoding the RBD-Plus of wild-type, Alpha and Beta variants fused to hFc with a larger linker (ADDDDK; “W1”, “E1” and “S1”) or a smaller linker (AAA; “W2”, “E2” and “S2”) (Figure 5A). The ELISA results showed that, whether a wild-type, Alpha or Beta sequence, fusion protein vaccines with a small linker produced higher IgG titers (Figure 5B). Among these groups, W2 antiserum maintained the highest antigen-specific IgG titer, which increased with time after immunization. The IgM titer in all groups also increased with time (Figure 5C). Notably, W2 antiserum elicited the highest NA titer, followed by S2 (Figure 5D). Meanwhile, we found that all subunit vaccines could induce the production of antibodies in mice sera and then effectively block the binding of RBD to ACE2, especially sera in mice immunized witth RBD-Plus linked with a smaller alanine linker significantly blocked the RBD binding to ACE2. (Figure 5E). This indicates that the RBD-Plus-hFc fusion protein vaccine with a small linker sequence can induce long-lasting antigen-specific antibody and neutralizing responses in mice.

### 3.6. SARS-CoV-2 Wild-Type RBD-Plus-hFc with a Small Linker Exhibits a High Titer of Neutralizing Antibodies against Variants Alpha, Beta, Delta and Omicron

The emergence of novel SARS-CoV-2 variants makes the development of a pan-coronavirus recombinant vaccine necessary. For a universal vaccine to be successful, it must provide a broad spectrum of antibody responses and lasting protection. Here, we show that a SARS-CoV-2 wild-type and Beta variant RBD fused with a human IgG1 Fc fragment with a small linker (“W2” and “S2”), elicited strong humoral-mediated immune responses and neutralizing antibody responses in immunized mice. To assess the cell-mediated immune responses of W2 and S2, we measured cytokine levels, including IL-4, IL-6, TNF-α and IFN-γ, in the antisera of immunized mice. As in Figure 6A,B, W2 and S2 immunization significantly increased antisera cytokine levels, especially IL-4, compared to the control. We next generated a panel of live wild-type SARS-CoV-2 variants, including Alpha, Beta, Delta and Omicron, to assess the neutralization breadth of antisera induced by the vaccine antigens. As shown in Figure 6C, a similar neutralization pattern was observed for W2 sera; specifically, the neutralizing antibody titles of anti-W2 sera against wild-type, Alpha, Beta, Delta and Omicron were 512, 512, 512, 512 and 256, respectively. Unexpectedly, compared to the Beta variant, wild-type recombinant protein showed stronger neutralizing titers against SARS-CoV-2 wild-type (fourfold), as well as Alpha (twofold), and Omicron (twofold) variants (Figure 6C).

## 4. Discussion

Fusion of the human IgG Fc region has many advantages when expressing a protein of interest, including binding to the Fc receptor on the surface of immune cells and enhancing a variety of immunological functions. Indeed, many therapeutic drugs use Fc-fusion proteins, which have no known side effects [26,27]. Therefore, it is safe to use the human IgG1 Fc region as the fusion partner for SARS-CoV-2 antigenic sequences [31]. In this study, we assessed six recombinant protein vaccines developed based on an RBD subunit with a human Fc or histidine tag and compared their ability to induce broadly neutralizing antibodies against the SARS-CoV-2 wild-type strain. Our results showed that, at the same antigen dose, the RBD-hFc vaccine was more potent than the RBD-His vaccine in eliciting long-lasting, high-titer broadly neutralizing antibodies in mice (Figure 2). At the same time, we demonstrated that RBD-hFc can stimulate both Th1 and Th2 immune responses (Figure 3 and Figure 4). Importantly, we show that the Fc-fusion protein can significantly enhance immunogenicity in vivo.

Linkers play an important role in the construction of stable, bioactive fusion proteins [28]. Linkers are amino acid sequences that serve to separate two fusion proteins and must be flexible enough to allow the proteins on either side to perform their independent functions [29]. Linker sequences can significantly increase the transfection efficiency of recombinant fusion protein [32]. We confirmed that a SARS-CoV-2 RBD-hFc protein with a small linker (AAA) elicited stronger humoral and cell-mediated immune responses compared to a longer linker (ADDDDK, Figure 5). Our study demonstrates that a recombinant SARS-CoV-2 protein vaccine with a 3-amino acid linker can induce strong antigenic activity in mice.

Despite some vaccine success, because SARS-CoV-2 is a low-fidelity RNA virus that relies on RNA polymerase, the high worldwide transmission rate provides an ideal scenario for the emergence of variants that are able to escape from antibody neutralization, such as B.1.1.7 (Alpha), B.1.351 (Beta), B.1.617 (Delta) and B.1.1.529 (Omicron) [33]. For SARS-CoV-2 immunization, protein subunit vaccines and nucleic acid vaccines are the most effective; however, studies have shown that newly emerged SARSCoV-2 variants can escape the neutralizing responses primed by mRNA vaccines [7]. In order to avoid reinfection, booster vaccinations must immunize against all strains. However, whether an ideal booster is the same as the original vaccine or, rather, a variant-specific vaccine is an issue that needs to be explored. We found that the wild-type SARS-CoV-2 RBD-Plus-hFc protein with an AAA linker exhibited a broader spectrum of antibody responses and provides better protection (Figure 5) compared to other vaccines. At the same antigen dose, the wild-type SARS-CoV-2 RBD-Plus-hFc immune sera had the highest IgG titer and levels of neutralizing antibodies. Excitingly, the antibodies produced in response to RBD-Plus-hFc showed good cross-neutralizing activity against wild-type, Alpha, Beta and Delta variants (Figure 6). Further, the RBD-Plus-hFc immune sera exhibited comparable neutralization potency towards the Omicron variant, despite the corresponding neutralizing titers decreasing by twofold relative to those against the wild-type strain. Hence, the present study demonstrates that the wild-type SARS-CoV-2 RBD-Plus-hFc recombinant protein vaccine can potently induce cross-variant neutralizing antibodies.

Cellular immunity helps destroy virally-infected cells and control the viral load in the body, while humoral immunity is responsible for clearing viral particles released by infected cells or through cellular immunity [34,35]. With the continued emergence of new SARS-CoV-2 strains, it is urgently necessary to develop a universal vaccine that can induce both strong cellular and humoral immunity [33]. Lymphocytes isolated from immunized mice showed an increased production of IFN-γ and IL-4 after stimulation with RBD-hFc (Figure 3C,D). Moreover, using flow cytometry, the number of the memory lymphocytes (including CD4^+^CD44^high^IL-4^+^IFN-γ^+^, and CD8^+^CD44^high^IL-4^+^IFN-γ^+^ cells) was found to be increased in the mice treated with RBD-hFc (Figure 4A,B). These data suggest that the RBD-hFc vaccine can induce both humoral and cellular immune responses in immunized mice.

The present data and our previous studies demonstrate that SARS-CoV-2 prototype stain-derived recombinant protein vaccines can not only strongly induce cellular and humoral immune responses to maximize immune protection, but also potently induce cross-variant neutralizing antibodies. These findings provide valuable information for the development of recombinant protein-based SARS-CoV-2 vaccines and support the continued use of approved recombinant protein SARS-CoV-2 vaccines in regions where variant viruses persist.

## Figures and Tables

**Figure 1 vaccines-10-01502-f001:**
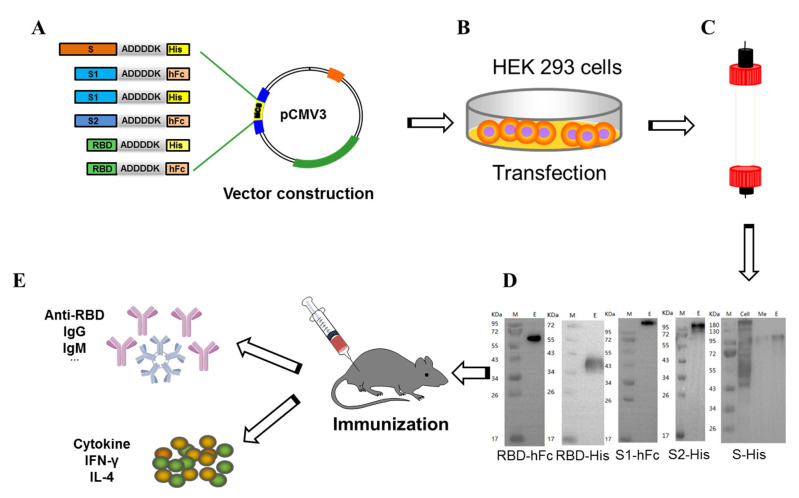
Schematic diagram of recombinant subunit protein vaccines and experimental design. Expression vectors containing either S-His, S1-His, S1-hFc, S2-His, plus an RBD-His or RBD-hFc fusion protein (**A**) was transfected into HEK293 cells (**B**). Recombinant fusion proteins were purified by column chromatography (**C**) and resolved and visualized by SDS-PAGE (**D**) (Lane M: protein marker; Lane E: elution target protein). (**E**) Schematic diagram of murine immunization with S-His, S1-His, S1-hFc, S2-His, RBD-His and RBD-hFc vaccines. Female Balb/c mice were immunized twice with S-His, S1-His, S1-hFc, S2-His, RBD-His and RBD-hFc fusion proteins (*n* = 5 per group).

**Figure 2 vaccines-10-01502-f002:**
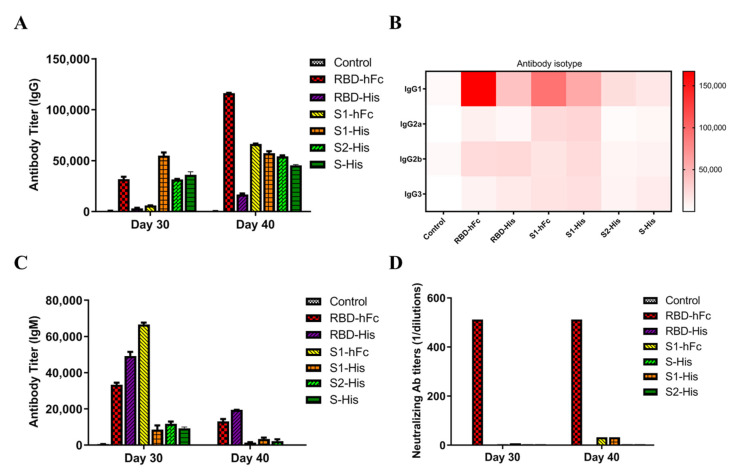
Serum-specific and neutralization antibody titers induced by recombinant vaccines. (**A**) Serum levels of IgG titer at 30 and 40 days after immunization were determined by ELISA. (**B**) Serum levels of IgG subtype titer at 30 and 40 days after immunization were determined by ELISA. (**C**) Serum levels of IgM titer at 30 and 40 days after immunization were determined by ELISA. (**D**) Neutralizing activity against SARS-CoV-2; data are presented as mean ± SD of absorbance at 450 nm (*n* = 5).

**Figure 3 vaccines-10-01502-f003:**
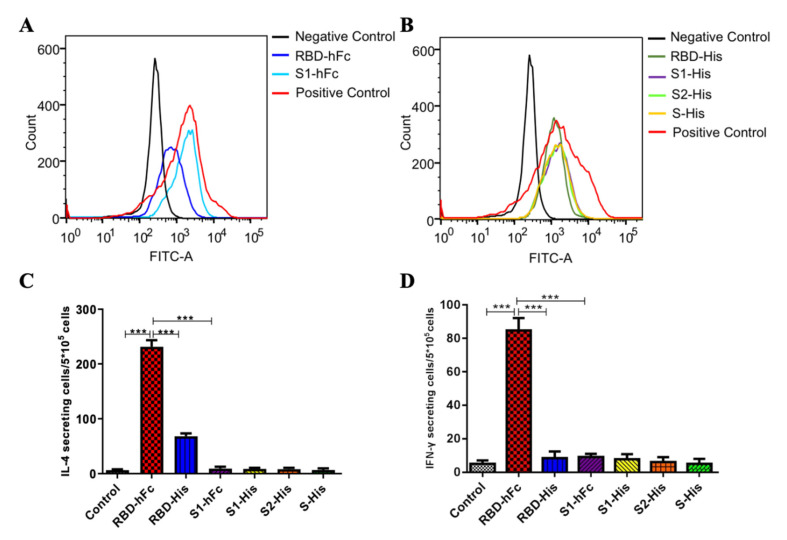
Mice antisera inhibited RBD binding to ACE2 and splenocytes of RBD-hFc immunized mice significantly increased cytokines secretion. (**A**) Inhibitions of RBD-His binding to cell-expressed ACE2 by RBD-hFc and S1-hFc mice antisera were measured by flow cytometry. (**B**) Inhibitions of RBD-hFc binding to cell-expressed ACE2 by RBD-His, S1-His, S2-His and S-His mice antisera were measured by flow cytometry. (**C**) Number of IL-4 secreting cells in splenocytes stimulated by recombinant subunit protein vaccines on day 40. (**D**) Number of IFN-γ secreting cells splenocytes stimulated by recombinant subunit protein vaccines on day 40. Data are presented as mean ± SD; *** *p* < 0.001.

**Figure 4 vaccines-10-01502-f004:**
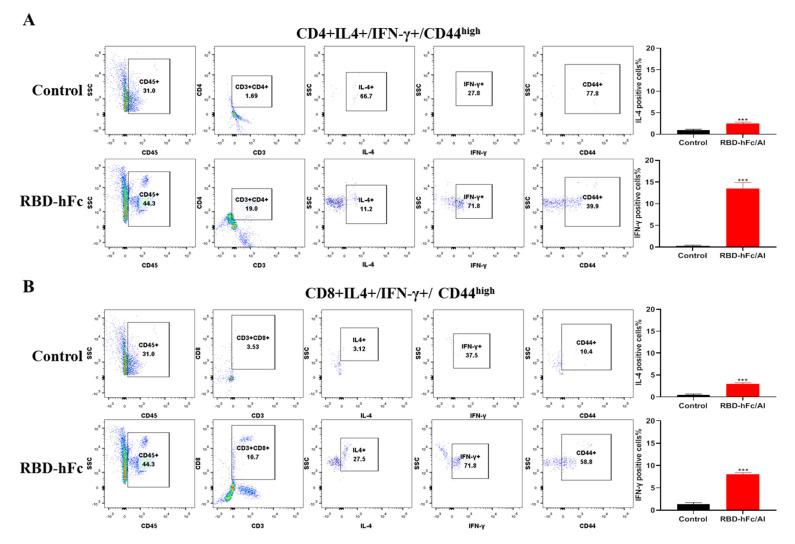
SARS-CoV-2 RBD-hFc vaccine induced a protective cell-mediated immune response in mice. The RBD-hFc fusion vaccine increases populations of memory CD4+ (**A**) and CD8+ (**B**) T cells. *** *p* < 0.001.

**Figure 5 vaccines-10-01502-f005:**
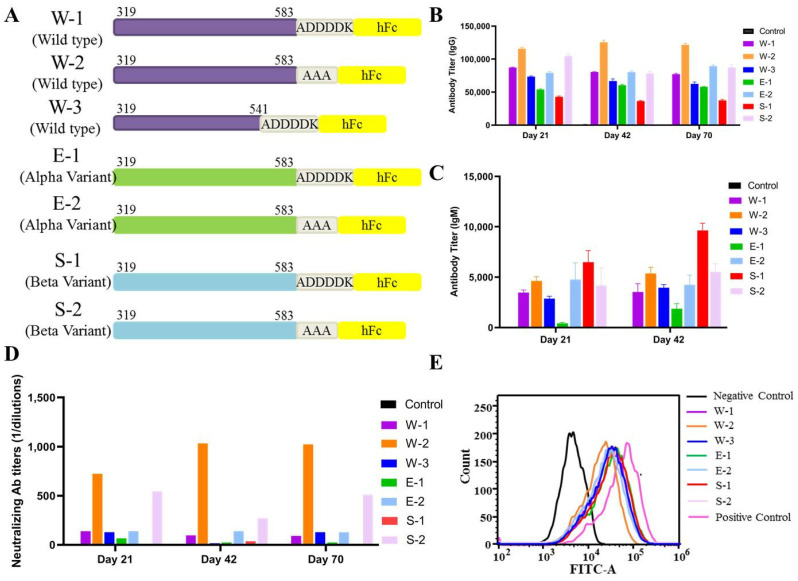
Evaluation of linker size on immunogenicity. (**A**) Expression vector sequence containing SARS-CoV-2 wild-type, Alpha and Beta RBD sequence fused to hFc with a larger or small linker. (**B**) IgG titers in mouse sera were determined by ELISA. (**C**) Serum levels of IgM titer on days 21 and 42 were determined by ELISA. (**D**) Neutralizing activity in response to the SARS-CoV-2, alpha or beta variants, challenge. (**E**) Flow cytometry shows inhibition of recombinant protein binding to ACE2.

**Figure 6 vaccines-10-01502-f006:**
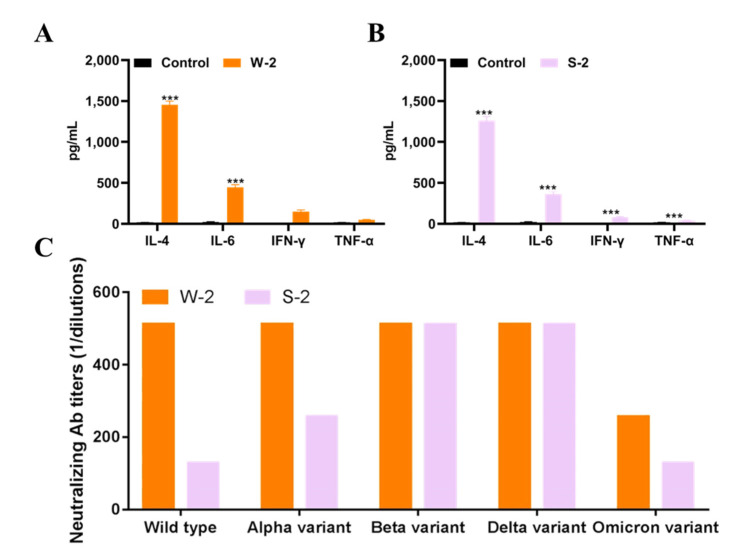
Cytokine and neutralizing antibody levels in immunized mice. IL-4, IL-6, TNF-α and IFN-γ levels in W2 (**A**) and S2 (**B**) mice were detected by ELISA. (**C**) Neutralization activities of W2 and S2 mice against wild-type, Alpha, Beta, Delta and Omicron variants were assayed by live virus. Data are presented as mean ± SD; ***, *p* < 0.001. All experiments were repeated at least twice.

## Data Availability

All data are available from the corresponding authors upon reasonable request.

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
