# Peer review of "Human Fc-Conjugated Receptor Binding Domain-Based Recombinant Subunit Vaccines with Short Linker Induce Potent Neutralizing Antibodies against Multiple SARS-CoV-2 Variants"

_vaccines, 2022, doi:10.3390/vaccines10091502_

Round 1
Reviewer 1 Report
In the submitted manuscript “Human Fc-conjugated receptor binding domain-based recombinant subunit vaccines with short linker induce potent neutralizing antibodies against multiple SARS-CoV-2 variants”, the author compared the immunogenicity between different RBD based recombinant subunit vaccines with different linker lengths. They also evaluated the cross-neutralization of different variants of SARS-CoV-2 with their wild type and mutant RBD based vaccines.
The manuscript is well drafted, and it can be published by addressing the following concerns.
1. The findings of the study are interesting but similar designs of RBD-FC based vaccines have been reported previously. The author needs to clarify what makes their design of these RBD-Fc vaccines novel
2. Also include the references of RBD-Fc based vaccines against SARS-CoV-2 reported recently.
3. Clearly state whether monomer or trimer of Spike proteins were used for vaccines design.
4. Change the statement line 229, These results show that RBD-hFc 228
5. elicits a robust neutralizing antibody response against live SARS-CoV-2 infection in mice. The vaccines haven’t been tested in challenge study in-vivo in mice.
6. RBD-Fc has higher antibody titer at day 40 compared to others. What attributes of this vaccines is responsible for this? Is it due to higher stability in-vivo or due to high antigenicity? Clarify.
7. Overall IgG titre is low in full length S-his compared to both S1 and S2. Can you speculate why this effect is observed.
8. Please clarify which antibody titre is more important in vaccine design. IgM levels observed is higher in S1-hfc vaccine at day 30 compared to RBD-hfc. What might be the reason?
9. For the neutralizing antibody titres, was the IgG purified or whole sera used? Is the neutralization a combination effect of both IgG and IgM present?
10. For the binding experiment of RBD to ACE-2, can you compare the inhibition between Fc tagged and his tagged vaccines.
11. Also provide a rational for selecting the particular linker used in this study. Did the author checked if the linker has some immunogenicity?
12. Also provide rational for design of WT-3 vaccine design?
13. It is quite interesting to see why beta1 vaccines with longer linker showed higher IgM titre than beta-2. Also the IgM titre are higher at day 42 compared to day 21 contrary to figure 2C. Please explain this.
14. Neutralizing activities of W2 vaccines is quite interesting against different variants of SARS-CoV-2. Can author put more light on this, this data is quite intriguing and can support that booster doses of Covid vaccines might be quite helpful in real scenario.
Author Response
- The findings of the study are interesting but similar designs of RBD-FC based vaccines have been reported previously. The author needs to clarify what makes their design of these RBD-Fc vaccines novel. Also include the references of RBD-Fc based vaccines against SARS-CoV-2 reported recently clearly state whether monomer or trimer of Spike proteins were used for vaccines design.
Response: Thank you so much for your insightful question. Indeed, many studies (PMID: 32225176, 32979942, 32155444) have proved that the RBD (319–541 amino acid region) is certainly considered a candidate for SARS-CoV-2 vaccine development. As a matter of fact, RBD monomer (PMID: 32726802, 32645327), RBD-homodimer (PMID: 34493710, PMID: 33731916) and RBD trimer (PMID: 34117252, 34179862) are being developed into vaccines. It is worth noting that the RBD-homodimer vaccine (zF2001) has been successfully approved in many countries. All these results indicate that RBD is a successful candidate for SARS-CoV-2 vaccine. Secondly, previous studies have shown that human IgG1 Fc fragment could not only enhance the vaccine immunogenicity, but also significantly improve conformation of RBD (PMID: 33731916, 33247109). Most importantly, we have been engaged in the development of RBD recombinant protein vaccines, and our previously study confirmed that RBD-mFc can significantly induce strong immune response in mice (PMID: 32613971). In this study, we continue to study RBD-hFc fusion recombinant protein vaccine, and found that Fc-fusion protein can significantly enhance immunogenicity in vivo. Meanwhile, we also performed immunogenicity comparison of RBD-hFc with different linkers, and confirmed that RBD-hFc protein with a small linker (AAA) elicited stronger humoral and cell-mediated immune responses compared to a longer linker. This is one of the first studies to directly compare how different linkers fusing the SARS-CoV-2 RBD domain might alter the immune responses in vivo. Furthermore, we found that SARS-CoV-2 wild-type recombinant protein showed stronger neutralizing titers against SARS-CoV-2 wild-type, as well as Alpha, Beta, Delta and Omicron variants. Our findings provide support information for continued use of approved recombinant protein SARS-CoV-2 vaccines in regions where variant viruses persist.
- Change the statement line 229, These results show that RBD-hFc 228 elicits a robust neutralizing antibody response against live SARS-CoV-2 infection in mice.The vaccines haven’t been tested in challenge study in-vivo in mice.
Response: Thank you so much for your knowledgeable question. Actually, the live SARS-CoV-2 virus challenge experiment was carried out in vitro by using the serum of immunized mice. The wrong expression has been revised and marked in red in the revised manuscript. Thanks again for your valuable and very helpful suggestion.
- RBD-Fc has higher antibody titer at day 40 compared to others. What attributes of this vaccine is responsible for this? Is it due to higher stability in-vivo or due to high antigenicity? Clarify.
Response: Thank you very much for your constructive question. As a matter of fact, Fc fragment can serve as an immunopotentiator to enhance the cellular and humoral immune responses to the vaccine since it facilitates antigen delivery and presentation through interacting with Fc receptors on antigen presenting cells. Meanwhile, the hFc-fusion can improve recombinant immunogen solubility and stability, and prolong immunogen half-life.Therefore, in our results, we found RBD-hFc triggered a robust humoral immune response in mice, followed by S1-hFc. Our results confirmed that RBD is a candidate for SARS-CoV-2 vaccine.
- Overall IgG titer is low in full length S-his compared to both S1 and S2. Can you speculate why this effect is observed.
Response: Thank you so much for your insightful question. In our study, we observed the order of IgG titers at day 40 was: RBD-hFc > S1-hFc ≈ S1-His ≈ S2-His ≈ S-His and there was no significantly different among S1-His, S2-His and S-His (P=0.349). This result is consistent with previously study that hFc can significantly enhance the immunogenicity of recombinant proteins in mice. At the same time, we used the same dosage of recombinant protein vaccines to immunized mice, as the the molecular weight of S protein was significantly higher than that of S1 and S2, the molar concentration of S-His was lower than that of S1-His and S2-His. Therefore, it is reasonable that IgG titer in S-His is low compared to both S1 and S2.
- Please clarify which antibody titre is more important in vaccine design. IgM levels observed is higher in S1-hfc vaccine at day 30 compared to RBD-hfc. What might be the reason?
Response: Thank you very much for your constructive question. It is widely recognized that IgM provide the first line of defense during viral infections. Therefore, the detection of IgM in the serum reveals a recent exposure to the virus, while the detection of IgG suggests that the exposure occurred several days before. Meanwhile, clinical results showed that a majority of COVID-19 patients developed both RBD-specific neutralizing antibody responses rapidly after SARS-CoV-2 infection. And the magnitude of RBD-specific IgG titers positively correlated with neutralization titers, the seroconversion of IgG or IgM was observed within 20 days after SARS-CoV-2 infection (PMID: 32221519, 32511565, 32311668). In our study, we observed the IgM level in S1-hFc vaccine at day 30 is higher than that of RBD-hFc vaccine.The might reason was that the conversion of IgM in S1-hFc immunized mice was less than in RBD-hFc immunized mice. Our IgG level confirmed the result, which was significantly lower in S1-hFc immunized mice serum than in RBD-hFc group at day 30.
- For the neutralizing antibody titres, was the IgG purified or whole sera used? Is the neutralization a combination effect of both IgG and IgM present?
Response: Very appreciated for your valuable question. As a matter of fact, the neutralizing antibody titres were detected by the pooled sera from each groups. It has been reported that higher titers of NAs indicate lower levels of virus replication and stronger protective capability against virus infection.
- For the binding experiment of RBD to ACE-2, can you compare the inhibition between Fc tagged and his tagged vaccines.
Response: Many thanks for your useful suggestion. Actually, in the binding experiment, RBD-His was used to compete with anti-RBD in sera from hFc-tag recombinant protein vaccined mice binding to ACE2, while RBD-hFc was contended with His-tag recombinant protein vaccined mice sera. Therefore, the binding experiment results in different recombinant protein vaccines were presented in different figure. The inhibition ratio was calculated in the same method as shown in the following figure 1. The order of inhibition efficiency is as follows: S1-hFc > S1-His > S2-His ≈S-His ≈RBD-His > RBD-hFc.
Figure 1. Histogram of average fluorescence intensity of recombinant subunit protein vaccines binding to cell-expressed ACE2 by mouse antisera was measured by flow cytometry
- Also provide a rational for selecting the particular linker used in this study. Did the author checked if the linker has some immunogenicity?
Response: Thanks very much for your constructive question. A generally acknowledged fact that Linker plays an important role in the construction of stable and biologically active fusion proteins. Excellent Linker is can not only significantly increase the transfection efficiency, but also flexible enough to allow protein to perform the independent conformation and function. It has been reported that compared to 13 amino acids, short linker (five amino acids) was the most effective for displaying the spike RBD (PMID: 33671255). In our study, we did not check the immunogenicity of linkers used in this study, which is generally not thought to produce immunogenicity in vivo because the linker is too short (3 or 6 amino acids). But your suggestion was very useful,the following experiments, we will consider testing the in vivo immunogenicity of the linker.
- Also provide rational for design of WT-3 vaccine design?
Response: Thanks very much for your insightful question. Actually, design of WT-3 vaccine was the same as the first part of the manuscript.In this section, WT-3 vaccine was just used to compare the immunogenicity of recombinant protein vaccines with different SARS-CoV-2 domain face to face.
- It is quite interesting to see why beta1 vaccines with longer linker showed higher IgM titer than beta-2. Also the IgM titre are higher at day 42 compared to day 21 contrary to figure 2C. Please explain this.
Response: Thank you so much for your valuable question. Indeed, no matter at day 21 and 42, IgM titer in beta1 vaccines with longer linker showed higher level compared to other groups. Nevertheless, relative to IgM, IgG levels were more remarkable, the IgM titer in beta1 vaccines with longer linker group is 10000, while IgG titer in wild-type vaccines with small linker is 100000, which is ten times higher than that of IgM titer. Meanwhile, wild-type vaccines with small linker antiserum elicited the highest NA titer. In summary, wild-type vaccines with small linker can induce long-lasting antigen-specific antibody and neutralizing responses in mice.
- Neutralizing activities of W2 vaccines is quite interesting against different variants of SARS-CoV-2. Can author put more light on this, this data is quite intriguing and can support that booster doses of Covid vaccines might be quite helpful in real scenario.
Response: Very appreciated for your insightful suggestion. It is truly interesting that SARS-CoV-2 wild-type recombinant protein showed stronger neutralizing titers against SARS-CoV-2 wild-type, as well as Alpha, Beta, Delta and Omicron variants. We are also very concerned about this result, because it can not only confirm the antigenic immunity of RBD recombinant protein vaccine in vivo, but also be effective against a variety of circulating variants, which provides a supporting basis for booster vaccination. We will continue to conduct further validation in other pre-clinical experimental animals, such as SD rats or monkey.

Reviewer 2 Report
Suggestions for authors
The manuscript titled “Human Fc-conjugated receptor binding domain-based recombinant subunit vaccines with short linker induce potent neutralizing antibodies against multiple SARS-CoV-2 variants” by Chen et al., presented recombinant SARS CoV-2 vaccine candidate with enhanced immunogenicity and broad neutralization activity.
The authors claimed that 1) fusion of human Fc to target protein enhances immunogenicity and is effective to induce both humoral and cellular immunity, 2) RBD319-583 with short linker to hFc trigger long lasting, high neutralizing antibodies in mice, 3) RBD319-583 of WA1/2020 strain with short linker to hFc induce broad neutralizing antibodies against variants and provided supporting results for each of these points.
There are few minor suggestions.
1. In line 61~63, “insufficient immune response due to its small size ~ makes it more susceptible to antigenic drift than larger recombinant proteins.” Please provide further explanation how the “insufficient immune response” increases the “susceptibility to antigenic drift”. To my knowledge, antigenic drift occurs due to genetic mutation during replication. Thus, mutation rate can be affected by size and stability of genome.
2. Line 64: Correct “simulate” to “stimulate”
3. Line 96: Add citation for WA1/2020 strain.
4. Line 111: Add spacing to “timepoint”.
5. Line 152: ELISpot “assays were”
6. Line 154: Provide material information if commercial mAbs against IL-4 and INF-r were used.
7. Line 159: Add spacing to “℃for”.
8. Line 189: Sub-title for the paragraph should represent the results of cloning vector construct for recombinant subunit protein vaccine antigen, their expression and purification, immunization.
9. Line 221 “Prototype vaccines”
10. Figure 1. D SDS-PAGE image is too small to read target size.
11. There are few results that do not correspond to Line 223~224. For instance, S1-his group’s IgG titer on day 30 is higher than S1-hFc (Fig 2A).
12. Figure 2. D Generally, neutralizing titers presented in log scale.
13. what is positive control for Fig 3B? It seems there is not peak shift between positive control and other experimental groups as described in line 244~245.
14. Please specify the description of control in Fig 5.
15. Line 330: Correct “neutralization” to “neutralizing antibody titers”
16. Line 342: Omit “sequence”
Author Response
- In line 61~63, “insufficient immune response due to its small size ~ makes it more susceptible to antigenic drift than larger recombinant proteins.” Please provide further explanation how the “insufficient immune response” increases the “susceptibility to antigenic drift”. To my knowledge, antigenic drift occurs due to genetic mutation during replication. Thus, mutation rate can be affected by size and stability of genome.
Response: Thank you so much for your knowledgeable question. This is our mistakes, as you mentioned antigenic drift occurs due to genetic mutation during replication, not the insufficient immune response. The originally meant is to express that the protein antigen is small in size and low in immunogenicity.These mistakes have been carefully corrected in the revised manuscript. Thanks again for your patience and carefulness on our manuscript.
- Line 64: Correct “simulate” to “stimulate”
Response: Thank you so much for your patience and carefulness on our manuscript. The detail incorrect spelling has been carefully corrected in the revised manuscript in red.
- Line 96: Add citation for WA1/2020 strain.
Response: Thank you so much for your constructive suggestion. Actually, all SARS-CoV-2 virus and its variants are supplied by Guangdong Center for Disease Control and Prevention (CDC). The detail information are shown in the following table, while the number of WA1/2020 strain is MN985325.1 (marked by red box).
[1]张欢,郑焕英,邹丽容,刘哲,梁丽君,彭晓放,张薇,柯昌文,武婕.广东省首例新型冠状病毒的分离与鉴定[J].病毒学报,2020,36(02):155-159.DOI:10.13242/j.cnki.bingduxuebao.003657.
- Line 111: Add spacing to “timepoint”.
- Line 152: ELISpot “assays were”
Response: Thank you so much for your great efforts in improving the quality of our manuscript. According to your detail reminding, these mistakes have been carefully corrected. Thanks again for your careful checking on our manuscript.
- Line 154: Provide material information if commercial mAbs against IL-4 and INF-r were used.
Response: Many thanks for your suggestion. Actually, the IFN-γ (2210402) and IL-4 (2210005) ELISPOT kits were were purchased from Dakewe Biotech Co., Ltd. These kits were pre-coated with high-affinity anti-IFN-γ or anti-IL-4 monoclonal antibody in PVDF plate. The IFN-γ and IL-4 secreted by splenocytes of vaccine immunized mice were detected according to the operation instructions. The information was marked in “RED” in the revised manuscript.
- Line 159: Add spacing to “℃for”.
Response: Thank you so much for your carefulness on our manuscript. The mistake has been corrected in the revised manuscript.
- Line 189: Sub-title for the paragraph should represent the results of cloning vector construct for recombinant subunit protein vaccine antigen, their expression and purification, immunization.
Response: Thank you so much for your constructive suggestion. The sub-title for the paragraph has been modified to "Construction, Expression, Purification, and Immunoassay of Recombinant subunit protein vaccines" in the revised manuscript. Thanks again for your patience and carefulness in improving the quality of our manuscript.
- Line 221 “Prototype vaccines”
Response: Thank you so much for your patience and carefulness on our manuscript. The mistake has been corrected in “RED” in the revised manuscript.
- Figure 1. D SDS-PAGE image is too small to read target size.
Response: Thank you so much for your patience and carefulness on our manuscript. The figure has been corrected in the revised manuscript.
- There are few results that do not correspond to Line 223~224. For instance, S1-his group’s IgG titer on day 30 is higher than S1-hFc (Fig 2A).
Response: Thanks very much for your patience and carefulness on our manuscript. The figure has been corrected in the revised manuscript.
- Figure 2. D Generally, neutralizing titers presented in log scale.
Response: Thanks very much for your valuable and knowledgeable suggestion. Actually, the neutralizing titers in mice serum completely suppressed 50% CPE of the wells were calculated by Reed-Muench method according to the reference (PMID: 17092615). As the dilution of mice sera was serial 4-fold dilutions (1:4 - 1:1024) as description in the “ Materials and Methods” section (line 167-168).
- What is positive control for Fig 3B? It seems there is not peak shift between positive control and other experimental groups as described in line 244~245.
Response: Many thanks for your useful suggestion. Actually, in the binding experiment, RBD-His was used to compete with anti-RBD in sera from hFc-tag recombinant protein vaccined mice binding to ACE2, while RBD-hFc was contended with His-tag recombinant protein vaccined mice sera. Therefore, the binding experiment results in different recombinant protein vaccines were presented in different figure. The inhibition ratio was calculated in the same method as shown in the following figure 1. The order of inhibition efficiency is as follows: S1-hFc > S1-His > S2-His ≈S-His ≈RBD-His > RBD-hFc.
Figure 1. Histogram of average fluorescence intensity of recombinant subunit protein vaccines binding to cell-expressed ACE2 by mouse antisera was measured by flow cytometry
- Please specify the description of control in Fig 5.
Response: Very appreciated for your constructive suggestion. Actually, as recombinant protein vaccines, adjuvants are usually required to stimulate the body to generate strong immune response. In this study, all vaccine preparations were prepared using aluminum as an adjuvant, while the control group was injection with same volume of PBS. These detail information were added in the "Materials and Methods" section (line 110-111) of the revised manuscript. At last, thank you so much for your patience in improving the quality of our manuscript.
- Line 330: Correct “neutralization” to “neutralizing antibody titers”
- Line 342: Omit “sequence”
Response: Thank you so much for your careful checking on our manuscript.According to your detail reminding, these mistakes have been carefully corrected. Many thanks for your great efforts in improving the quality of our manuscript.
